# Thermal assisted self-organization of calcium carbonate

Gan Zhang[1,2], Cristobal Verdugo-Escamilla [1], Duane Choquesillo-Lazarte [1] & Juan Manuel García-Ruiz [1]

Fabrication of mineral multi-textured architectures by self-organization is a formidable challenge for engineering. Current approaches follow a biomimetic route for hybrid materials based on the coupling of carbonate and organic compounds. We explore here the chemical coupling of silica and carbonate, leading to fabrication of inorganic–inorganic biomimetic structures known as silica-carbonate biomorphs. So far, biomorphic structures were restricted to orthorhombic barium, strontium, and calcium carbonate. We demonstrate that, monohydrocalcite a hydrous form of calcium carbonate with trigonal structure can also form biomorphic structures, thus showing biomorphic growth is not dictated by the carbonate crystal structure. We show that it is possible to control the growth regime, and therefore the texture and overall shape, by tuning the growth temperature, thereby shifting the textural pattern within the production of a given architecture. This finding opens a promising route to the fabrication of complex multi-textured self-organized material made of silica and chalk.

[1] Laboratorio de Estudios Cristalográficos, Instituto Andaluz de Ciencias de la Tierra (CSIC-UGR), Avenida de las Palmeras 4, E-18100 ArmillaGranada, Spain.
[2] Department of Structural Biology, Weizmann Institute of Science, 76100 Rehovot, Israel. Correspondence and requests for materials should be addressed to J.M.Gía-R. (email: juanmanuel.garcia@csic.es)

Calcium carbonate is the material most used by living organisms to fabricate complicated architectures that—upon selection by evolution—are used as efficient exoskeletons, optical devices, and other functions[1]. To make them, all that life needs is calcium from the sea or from fresh natural water, $CO_2$ from the atmosphere, and polymeric organic matter provided by the organisms. Most biomineral architectures are extremely complex in morphology and they are in many cases multi-textured, i.e., different textures and even mineral phases are produced in the same biomineral. This ability of living organisms to engineer hierarchical hybrid nanocomposites is evidently worth emulating in designing and producing novel materials. Most of the intense work devoted in the last years to synthetize biologically inspired textures and shapes follows the hybrid route found by life, i.e., using organic polymer to guide the nanocrystallization of calcium carbonate[2–4]. Recent advances have demonstrated the preparation of bilayered composites with different textural organization[5], the production of thin film mimicking the prismatic layer of biomineral $CaCO_3$, and the synthesis of nacre layers of shells[6–8], and polymer/clay nanocomposites[9]. However, the production of different coexisting textures during the same process requires a multi-step recipe with much external control. While the advances in this field are meritorious, this top-down approach still needs a high degree of external information to guide fabrication.

There is also a less explored but promising non-hybrid inorganic–inorganic route to synthesize complex self-organized hierarchical composite materials. It has been demonstrated that polymeric silica (playing a role similar to that of organic polymers in biominerals) and carbonate can self-organize into inorganic–inorganic complex architectures. They are made of millions of carbonate nanocrystals that are co-oriented via silica-induced interactions into a variety of biomimetic textures and morphologies[10–13]. They are named "biomorphs" because (a) their morphologies are reminiscent of the shape of primitive living organisms[14,15], and (b) the carbonate nanocrystals self-arrange forming complex textures from nanometer to millimeter scale that are also reminiscent of biominerals and biomimetic hybrid organic/inorganic composites[16,17]. Beyond the applications of silica biomorphs to functionalization pathways[18,19], and their potential for tissue regeneration and photonic

microarchitectures[20,21], the fascinating diversity of textures found in silica-carbonate biomorphs open many possibilities for the self-organized production of complex materials. So far, silica-carbonate biomorphs have been synthetized only with crystalline orthorhombic alkaline earth carbonates, namely witherite ($BaCO_3$), strontianite ($SrCO_3$), and aragonite (orthorhombic polymorph of $CaCO_3$) (Fig. 1). We have performed a screening of initial conditions seeking for the precipitation of calcium carbonate in alkaline silica gels by counterdiffusion method. We found that calcite and aragonite can concomitantly precipitate in time and space at certain initial concentrations of $CaCl_2$ [0.2 M] and $Na_2CO_3$ [0.2 M], highlighting the different textural and morphological behavior of both phases[22]. The calcitic architectures keep the crystal symmetry of trigonal structure of calcite, and behave optically like a single crystal at the scale of visible light and X-rays, while the aragonitic ones display complex non-crystallographic shapes and are clearly polycrystalline[22]. The failure of calcite's trigonal structure to fabricate biomorphs has been interpreted to mean that the orthorhombic crystalline structure of the carbonate phase was crucial for making complex non-crystallographic structures.

However, by changing the initial temperature and the concentrations of $CaCl_2$ and $Na_2CO_3$ to [0.05 M] we have found that monohydrocalcite, a hydrous form of calcium carbonate ($CaCO_3 \cdot H_2O$, herein MHC) with trigonal structure can also form biomorphic structures. When the screening was extended to explore the effect of temperature we found that in addition to classical spherulites and fiber-like crystals, MHC exhibits between 45 and 60 °C complex shapes and self-organized textures characteristic of biomorphs (Fig. 2)[23]. The rationale behind the exploration of the effect of temperature is based on a very fundamental aspect of biomorphs. The formation of biomorphs is an autocatalytic phenomenon triggered by the reverse solubility of calcium carbonate and silica on pH: as the carbonate crystals form, the pH decreases due to the continuous removal of carbonate groups, and this decreasing pH induces the precipitation of amorphous silica; in turn, the precipitation of silica causes an increase in pH and therefore triggers a new event of carbonate nucleation, thus maintain this autocatalytic cycle of co-precipitation[11,23]. Interestingly, this reverse behavior is stressed with temperature (Supplementary Fig. 1): the solubility of MHC

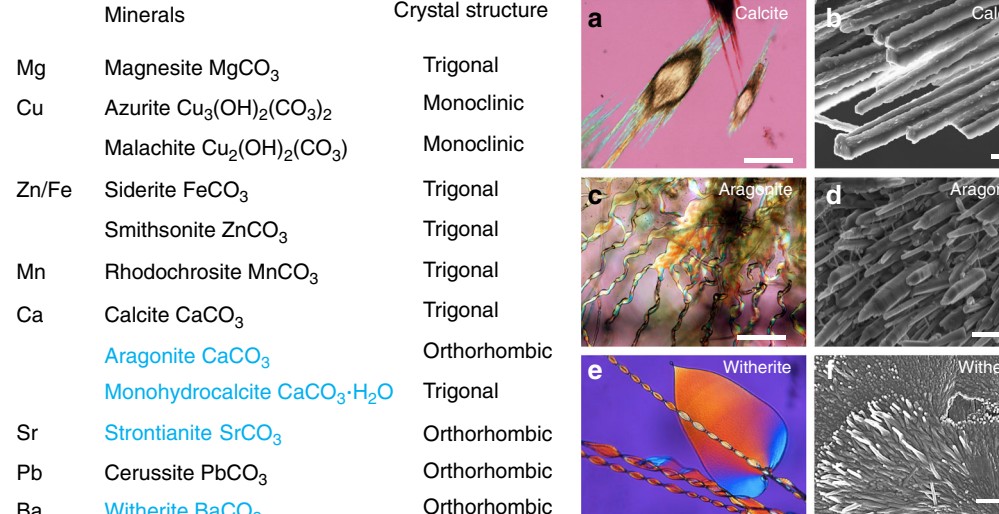

| | Minerals | Crystal structure |
|---|---|---|
| Mg | Magnesite $MgCO_3$ | Trigonal |
| Cu | Azurite $Cu_3(OH)_2(CO_3)_2$ | Monoclinic |
| | Malachite $Cu_2(OH)_2(CO_3)$ | Monoclinic |
| Zn/Fe | Siderite $FeCO_3$ | Trigonal |
| | Smithsonite $ZnCO_3$ | Trigonal |
| Mn | Rhodochrosite $MnCO_3$ | Trigonal |
| Ca | Calcite $CaCO_3$ | Trigonal |
| | Aragonite $CaCO_3$ | Orthorhombic |
| | Monohydrocalcite $CaCO_3 \cdot H_2O$ | Trigonal |
| Sr | Strontianite $SrCO_3$ | Orthorhombic |
| Pb | Cerussite $PbCO_3$ | Orthorhombic |
| Ba | Witherite $BaCO_3$ | Orthorhombic |

**Fig. 1** Mineral phases of carbonate and their corresponding crystal structures. The calcite (**a**, **b**), aragonite (**c**, **d**), and witherite (**e**, **f**) formed at silica-rich alkaline conditions are shown in optical micrographs, and the corresponding textures are illustrated by FESEM images. So far, complex biomorphic structures were restricted to orthorhombic $BaCO_3$, $SrCO_3$, and $CaCO_3$ (aragonite) carbonates. We have found that biomorphs of the trigonal phase MHC can be also synthetized under certain range of temperature. Scale bars: 100 μm (**a**), 200 μm (**c**), and 1 μm (**b**, **d**, **f**)

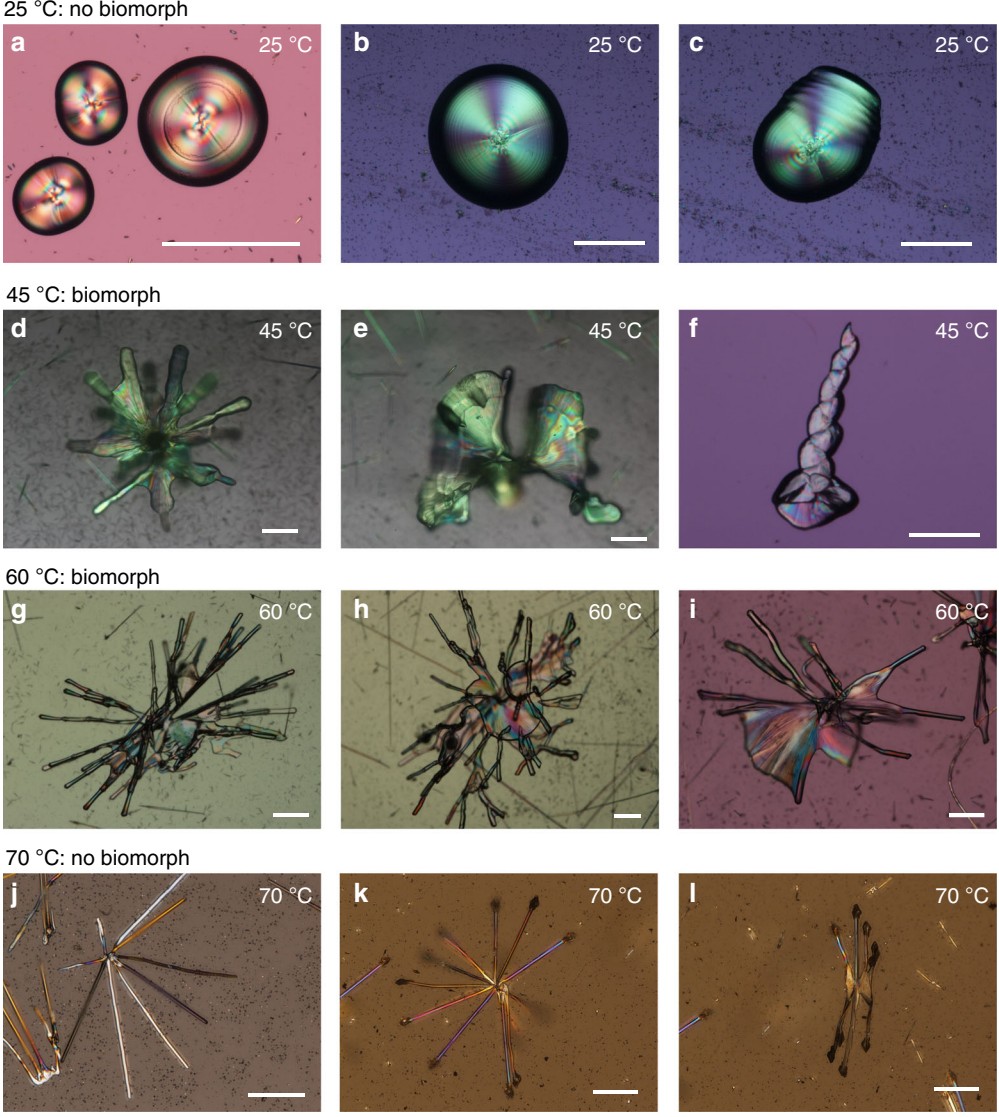

**Fig. 2** MHC aggregates grown at different temperatures. Optical micrographs of hemispherical (**a**, **b**) and caterpillar-like (**c**) aggregates at 25 °C; flower-like (**d**, **e**) and twisted ribbon-like (**f**) aggregates at 45 °C; curvilinear sheet and filamentary aggregates at 60 °C (**g–i**); star-like aggregates at 70 °C (**j–l**). Scale bar: 200 μm

decreases with temperature[24], while the solubility of silica increases with temperature[25–27].

Therefore, by screening temperature on the crystallization of monohydrocalcite we touch the coupling of silica and calcium carbonate supersaturation as well as the silica condensation. We show that the different coupling of MHC and $SiO_2$ precipitation affects the overall shape and the texture of the forming MHC-silica composites at specific ranges of temperature. We have used this finding to demonstrate the controlled production of complex self-organized biomimetic microarchitectures.

## Results and Discussion

**Temperature generated structures**. The crystallization of MHC is performed at different temperatures in alkaline silica gel by counterdiffusion method[28,29]. The carbonate bearing silica gel is set at pH $10.5 \pm 0.1$ within the lower part of a custom-made glass cassette (Supplementary Fig. 2 and Methods section). The calcium chloride solution is injected to the upper part of the cassette to start the counterdiffusion of reactants. The precipitation of MHC takes place in solution, upon diffusion of the $HSiO_4^-$,

$SiO_4^{2-}$, and $CO_3^{2-}$ species forming at these pH values, from the gel to the solution. The precipitated mineral phase is characterized by X-ray diffraction and in situ Raman microspectroscopy. It is shown that MHC is the first crystalline phase to nucleate and that it is stable in time (Supplementary Fig. 3 and Supplementary Fig. 4). Figure 2 shows optical micrographs of the MHC precipitates obtained at 25, 45, 60, and 70 °C. We can distinguish four types of mineral architectures. At room temperature, MHC forms the polycrystalline spherulites classical of this mineral phase (Fig. 2a, b) and caterpillar-like structures (Fig. 2c)[29,30]. At 45 °C, MHC aggregates are composed of continuous laminar sheets (larger than 500 μm in width) resembling flower-like structures (Fig. 2d, e). The sheets eventually experience curling to form twisted ribbons (Fig. 2f). These remarkable shapes, which arise from the counter propagation of two approaching curls, are considered a most typical feature of silica-carbonate biomorphs (Supplementary Fig. 5)[11]. At 60 °C (Fig. 2g–i) the growth rate of the counter propagating curls is faster than the radial growth of the sheets. As a result, the curvilinear sheets are stylized and most of them become cylindrical branches and form long filaments. The branches do not grow continuously, but they display a

bamboo-like structure with joints along the growing vector (Supplementary Fig. 6). These joints most likely formed during microscopic observation performed at room temperature. Finally, the experiments performed at 70 °C yield non-biomorphic MHC (Fig. 2j–l). They produce single crystals and crystal aggregates with morphologies controlled by the trigonal symmetry of the MHC crystal structure. This is most probably due to the high solubility of silica at this high temperature (Supplementary Fig. 1 and Supplementary Fig. 7) which prevent the co-precipitation with MHC, so that the growth of MHC is under the control of its crystal structure. These results demonstrate for the first time that the ability of a mineral phase to self-organize into nanocomposites with non-crystallographic morphologies is not privative of the orthorhombic structure of classical $BaCO_3$, $SrCO_3$, and $CaCO_3$ (aragonite) as previously reported[12,13,31]. This discovery opens new opportunities in finding other materials and crystal phases that also would produce biomorphs. It also suggests that a full understanding of texture formation and morphogenesis should focus on particle shape and interparticle bonding of supramolecular building blocks rather than on the symmetry of the crystalline structure.

**Complex structures created using temperature increases.** We have taken advantage of the effect of temperature on the texture and morphology of MHC to explore for first time the manufacture of heterostructured complex shapes of calcium carbonate by self-organization. After setting the silica gels, the cassette and the calcium chloride solutions are stored at the starting nominal temperature (25, 45, 60, or 70 °C). Then, the calcium solution is injected on top of the gel to begin the counterdiffusion experiment. The modulation of temperature is achieved by stepwise adjustment from one temperature to other. Different sequences of temperatures and corresponding residence times are selected to intend the controlled production of biomimetic structures. In particular, sea urchin-like MHC formed at the sequence of 25 and 70 °C (Fig. 3); the complex heterotextured MHC formed at the sequence of 25, 45, and 70 °C (Fig. 4), and the more complicated microarchitectures of MHC formed at the sequence of 25, 70, 45, 60, and again 70 °C (Fig. 5).

Figure 3 shows a simple route, starting at 25 °C for two days and then storing the experimental set in a thermostated oven previously heated at 70 °C. The growth process is followed by optical microscopy and is shown in Fig. 3a and Supplementary Fig. 8. At 25 °C spherulitic MHC formed (Fig. 3b). During the heating, the experiments move across the temperature range 45–60 °C for about 30 min, during which the spherulites experience the growth of a polycrystalline thin layer that, in most cases, cover their whole surface, and is the substrate for the formation of polycrystalline clusters. Once the nominal temperature of 70 °C is reached, the growth regime changes and MHC faceted sticks grow from the polycrystalline clusters of the spherical particles (Fig. 3d and Supplementary Fig. 8). The final architecture is a sea urchin-like shape consisting of different coexisting self-organized textures (Fig. 3c, e, g, h). Specifically, the spherulites formed at 25 °C exhibit a multilayered texture (Fig. 3c); while the sticks formed at 70 °C exhibit faceted textures characteristic of trigonal crystallographic symmetry (Fig. 3e, g, h). These results (Fig. 3 and Supplementary Fig. 9) prove that heterotextured complexity can be achieved by temperature control.

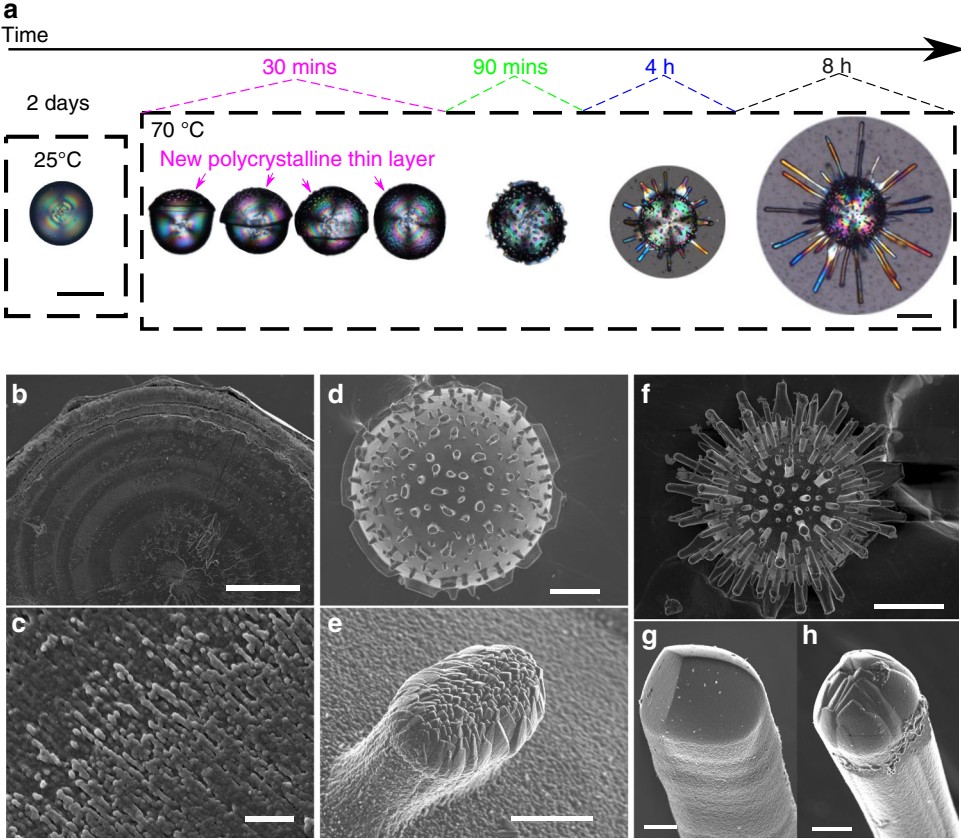

**Fig. 3** Growth process of a sea urchin-like MHC architecture. **a** Optical micrographs showing the evolution of morphology during the growth process of MHC under the stepwise modulation of temperature (25 and 70 °C); **b**–**h** selected FESEM images of MHC particles and corresponding textures formed at different stages: spherulites at 25 °C (**b**, **c**), early stage of sea urchin-like particle (**d**) and the developing faceted stick (**e**) at 70 °C, and final shape of sea urchin-like particle (**f**) and the faceted sticks (**g**, **h**) at 70 °C. Scale bars: 50 μm (**b**, **d**), 1 μm (**c**), 100 μm (**a**, **f**), 5 μm (**e**, **g**, **h**)

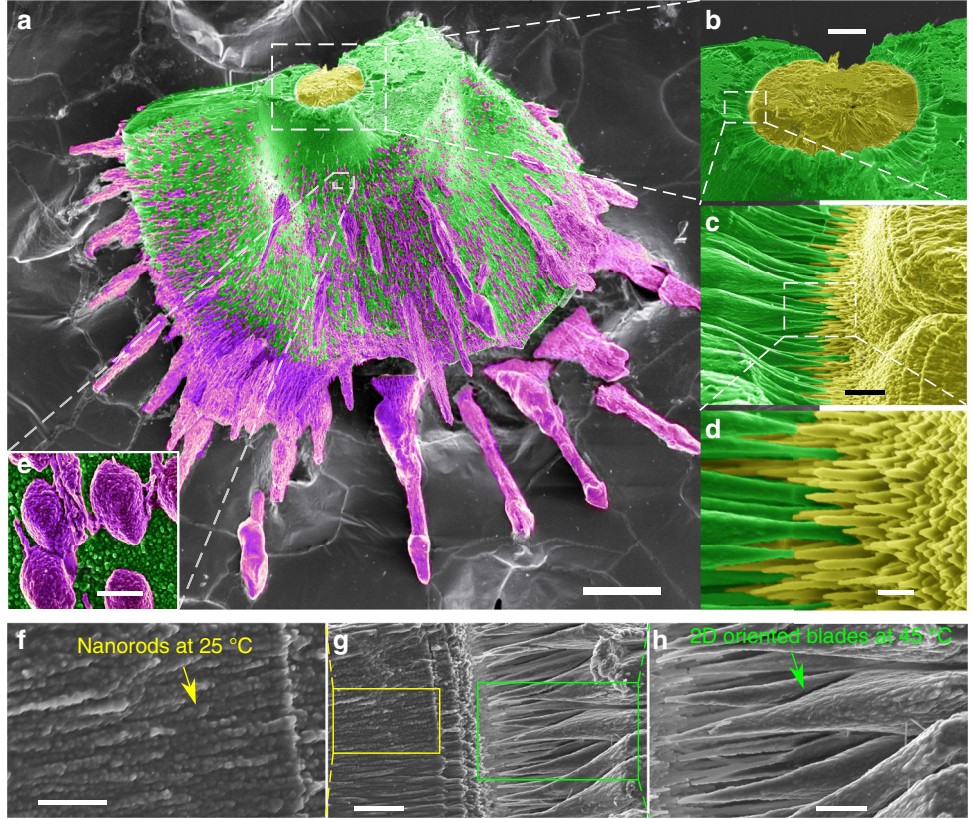

**Fig. 4** Heterotextured complexity of MHC. The growth periods at different temperatures have been labeled with false colors (**a**): 24 h growth at 25 °C (yellow), 6 days growth at 45 °C (green), 24 h growth at 70 °C (magenta), successively. The close-up views (**b**–**d** and **f**–**h**) show the transition from the spherulitic core (yellow) grown at 25 °C to the biomorphic laminar sheet (green) grown at 45 °C. The close-up view with higher magnification (**d**) shows the gradual transition of the textures from micron-size long rods of one hundred nanometer in thickness (25 °C, yellow) to 2D oriented blades (45 °C, green); close-up view (**e**) shows the mesocrystal with crystallographic symmetry made by nanodrops aggregation on the surface of laminar sheet at 70 °C (magenta). Close-up views (**f**–**h**) shows further details of the transition from nanorods to blades and also show that both, the nanorods and the blades are made by accretion of nanodrops. Scale bars: 100 μm (**a**), 20 μm (**b**), 2 μm (**c**, **g**), 500 nm (**d**), 1 μm (**e**, **f**, **h**)

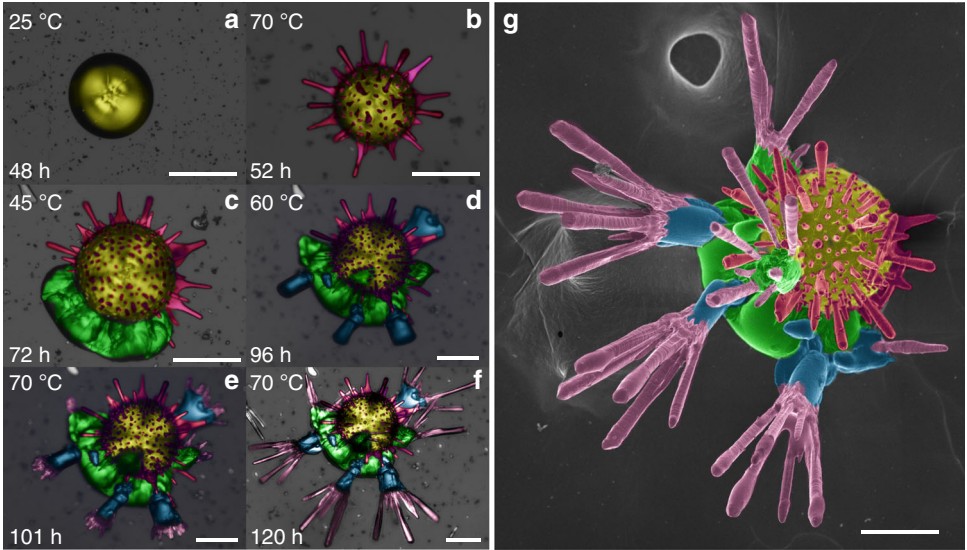

**Fig. 5** Growth history of multi-textured complex architectures of MHC. The growth periods at different temperatures have been labeled with false colors in both optical micrographs (**a**–**f**) and FESEM image (**g**): 48 h growth at 25 °C (yellow), 4 h growth at 70 °C (magenta), 20 h growth at 45 °C (green), 24 h growth at 60 °C (blue), 24 h growth at 70 °C again (purple), successively. The multi-textured MHC experience the growth at 25 °C (0–48 h), 70 °C (48–52 h), 45 °C (52–72 h), 60 °C (72–96 h), and again 70 °C (96–120 h). Scale bar: 100 μm

Figure 4 shows a more complex modulation of temperature. The experiment starts at 25 °C for 24 h, and then the temperature successively changes to 45 °C for 6 days and 70 °C for another 24 h. The final architecture (Fig. 4a) is made of three different textures, which can be used as the proxy to represent the corresponding temperatures at the growth history (Fig. 4). The transition between textures is rather smooth because in all cases the growth takes place by accretion of nanoparticles and not by classical ion-to-ion mechanism[32]. Whether the nanoparticles are either amorphous or dense liquid drops made of calcium carbonate and silica is unknown. The nanoparticles self-aggregate into nanorods with MHC structure. These nanorods spontaneously self-organize into different growth textures depending on temperature, i.e., on the concentration of species of silicate and carbonate. The self-assembly mechanism leading to different textures is likely driven by intrinsic anisotropic dipole–dipole interactions between the nanorods[33], but the details need to be explored. For instance, the core of the architecture in Fig. 4 is a spherulite made of independent micron-sized long rods with radial orientation formed at 25 °C (Fig. 4b); at 45 °C, each of the long rods transform into two-dimensional oriented blades which are also composed of the nanodrops (Fig. 4f, g, h), and these oriented blades merged each other to further construct the biomorphic laminar structures (Fig. 4c, d); at 70 °C, MHC nanoparticles self-assemble into trigonal faceted aggregates with trigonal crystallographic symmetry (Fig. 4e).

**Structures created by increases and decreases in temperature.** The reversibility of the temperature control is tested with a more complex thermal route (Fig. 5 and Supplementary Fig. 10). The experiment is set at 25 °C for 2 days to form MHC spherulites (Fig. 5a, yellow). Later, the temperature is increased to 70 °C until the sea urchin-like shapes are obtained after 12 h at that temperature (Fig. 5b, magenta). Then, the temperature was set back to 45 °C to allow the two-dimensional growth of curved sheets under biomorphic regime for 20 h (Fig. 5c, green). Then, the temperature is increased to 60 °C, at which the sheets experience a transformation to cylindrical branches during 24 h (Fig. 5d, blue), as explained in the supporting information (Supplementary Fig. 5). Finally the temperature is set at 70 °C again to fabricate a final phase made of elongated MHC crystals (Fig. 5e, f, purple). The whole process produces the multi-textured complex architectures of MHC shown in Fig. 5g.

Our results demonstrate that self-organized heterotextured architectures of calcium carbonate can be fully controlled with high level of complexity and precision. The morphology and self-arrangement of the carbonate growth units can be tuned by a simple control of temperature, thus controlling the time sequence of textures and the overall shape of the product. Besides, the finding that MHC may form either complex hierarchical biomorphic architectures or simple spherulites as a function of temperature, another important conclusion of these experiments is that the crystal structure of the mineral phase is not crucial to synthetize silica/carbonate biomorphs. The discovery that the formation of biomorphic structures is not restricted by crystal structure further suggests that the inorganic–inorganic route to the production of complex self-assembled materials is much more promising than previously expected.

## Methods
**Crystallization of monohydrocalcite**. The crystallization of monohydrocalcite is performed by counterdiffusion method in a glass cassette made of two rectangular glass plates separated by a rubber frame that works as a spacer, providing an inner space with dimension at 100 mm × 50 mm × 2 mm (Supplementary Fig. 2). The cassette is half filled with a silica gel at pH 10.5 containing 0.05 M Na₂CO₃ in concentration. After gelling, a 0.05 M CaCl₂ solution is injected in the cassette on top of the gel to start the counterdiffusion. The Na₂CO₃ bearing gel is prepared by dissolving 1.39 g sodium silicate (Sigma-Aldrich, reagent grade, Ref. 338443, replaced monthly) in 9 mL 0.05 M sodium carbonate solution (Na₂CO₃, ≥ 99.0%, Sigma-Aldrich), and then acidified by adding 3.5 mL of 1 M hydrochloric acid solution (analytical reagent, Fluka). The calcium chloride stock solution (CaCl₂, ≥ 99.0%, Sigma-Aldrich) is prepared at room temperature and then heated to the desired temperature. Purified water with an electrical conductivity <10⁻⁶ S m⁻¹ is used in all experiments. Before the injection, the Na₂CO₃ bearing gel and the CaCl₂ solution are placed in a thermostated oven beforehand to ensure the desired temperatures of crystallization. For the stepwise adjustments of the temperatures, the thermostated ovens are also previously set at desired temperature, and then the crystallization cassettes are transported to the oven to limit the gradual change in the temperature. The crystallization process is optically monitored by using a Nikon AZ100 optical microscope. Each observation is done in < 5 min to guarantee minimum temperature fluctuations. The obtained crystals are coated with carbon and examined by a field emission scanning electron microscope (FESEM) with an AURIGA system (Carl Zeiss SMT).

The counterdiffusion of Ca²⁺ and CO₃²⁻ provokes the chemical gradient along both the liquid part and the gel part in the crystallization cassette. Hemispherical monohydrocalcite forms in the liquid part, where Ca/CO₃ ratio is higher than 1; while elongated sheaf-of-wheat like calcite forms in the gel part, where the Ca/CO₃ ratio is lower than 1.

**Characterization of monohydrocalcite at different temperatures**. After crystallization, the resulting biomorphic precipitates are extracted out of the growth cassette and then analyzed by X-ray diffraction (Bruker D8 Venture, Cu Kα). In all cases and for all temperatures the precipitates are composed of monohydrocalcite (Supplementary Fig. 3). The crystallization at 45 and 60 °C are also analyzed by time-lapse measurements at different growth stages (12, 24, and 48 h) by in situ Raman microspectroscopy with a wavelength of 532 nm (LabRAM-HR spectrometer, Jobin-Yvon, Horiba, Japan). All the spectra collected from the biomorphic aggregates under these temperatures exhibit clear bands at 696, 719, and 1067 cm⁻¹ (Supplementary Fig. 4), which correspond to the characteristic peaks of the carbonate group in monohydrocalcite. In addition, the sea urchin-like heterotextured architectures formed by stepwise adjustment of temperature of growth (25–70 °C) are also identified as monohydrocalcite by X-ray diffraction (Supplementary Fig. 3).

## Data availability
The authors declare that the data supporting the findings of this study are available within the paper and its Supplementary Information files or from the corresponding authors upon reasonable request.

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

## Acknowledgements

The research leading to these results has received funding from the European Research Council under the European Union's Seventh Framework Program (FP7/2007-2013)/ ERC grant agreement n° 340863 "Prometheus". G.Z. acknowledges the Spanish Consejo Superior de Investigaciones Científicas for the pre-doctoral fellowship, within the program "Junta para la Ampliación de Estudios". C.V.-E. acknowledges the Spanish MINECO for contract PTA2015-11103-I. The authors acknowledge Alicia González Segura and Isabel Guerra-Tschuschke from Center of Scientific Instrumentation of the University of Granada for their technical assistance.

## Author contributions

G.Z. designed and performed the experiments, analyzed the data and discussed the results; C.V.-E. and D.C.-L. performed the X-ray studies; J.M.G.-R. discussed the results and wrote the paper.

## Additional information

**Competing interests:** The authors declare no competing interests.

