## [Peer Review File · Nature Communications]

Reviewers' comments:

Reviewer #1 (Remarks to the Author):

The authors report a rich self-assembly pathway of trigonal-monohydrocalcite (MHC) based growth of biomorphs as well as spherulitic and filamentary crystals as a function of temperature. The temperature control over the textures and overall morphologies allows for the targeted growth of hierarchical complex structures that modulate the shape and the composition. In my opinion, these are novel results to be published at the level of Nature Communications. The structures are very well characterized given the complexity of the emergent shapes. Before I recommend publication, some minor points mainly with regards to a clearer presentation of the results could be addressed:

1- Lines 69-77 very clearly demonstrate what is new in this manuscript; however, the abstract, although listing novel results, does not clearly communicate the improvements that this manuscript reports over the previous literature. I believe the abstract must be improved along these lines.

2- Lines 85-87: Can the authors provide a brief explanation or a qualitative diagram of the anticipated solubility rates of MHC and silica? After all, the statement as it is does not explain why biomorphs do not emerge at 25°C and only spherulites grow.

3- Lines 120-121: The geometry described here is not clear without the aid of visuals: do the authors have a movie or zoomed-in images of Figs. 2 (g-i)? Or a schematic that helps the reader as to what is a "bamboo-like structure," where are the joints? And arrows depicting the growth vector on these figures?

4- Line 124: The authors state that the morphologies are controlled by the trigonal symmetry of MHC. However, lines 130-133 and 234-237 oppose that, which I agree with. Perhaps the authors do not see any inconsistency; in any case the statement on Line 124 should be rephrased/clarified.

5- Lines 147-149: Can the authors put a reference to the corresponding figures to guide the reader?

6- Line 154: The curving laminar sheet is not clear in Fig. 3A, as well as its plane: is the surface normal in the z-axis of the figures?

7- Line 189 and Fig 4a: Is the green region a curved 2D laminar sheet? How do each of the long rods transform into this geometry? Do the authors mean that the valleys within the rods are deposited with silica, and a sheet in turn emerges? Or are there multiple 2D sheets that exhibit a laminar superstructure at the edge-on view; and hence not visible in Fig. 4a? A more detailed explanation and/or some more figures or some guidance (text labels, arrow etc.) added to Fig. 4a would greatly help.

8- Line 201: I suppose that the authors meant "nanometers," not "micrometer."

9- The paragraph starting at line 206: Can the authors put the words yellow, red, green, blue, and pink to the text to enable the reader interpret the figure directly from the text?

I believe that these remarks will improve the presentation of the manuscript and not only make it more interesting for experimentalists, but also for the theorists concerned with the thermodynamics and morphogenesis of biomorph growth, thereby covering a wide-range of readership as required for a Nature Communications publication.

Reviewer #2 (Remarks to the Author):

The manuscript by Zhang et al. reports the growth of biomorphs consisting of CaCO₃ and silica in which complex morphologies are controllably modulated via changes in solution temperature. This is the first report of biomorph growth in a non-orthorhombic crystal system and the first to show that temperature can be used as the control parameter in directing the morphology and in building morphological complexity. Biomorphs represent a particularly significant form of biomimetic structures, because their generation does not require introduction of macromolecular scaffolds, bioorganic additives, or compartmentalization. Rather simple physical controls resulting from coupled precipitation-pH oscillations are all that is required to generate these complex crystalline forms. Consequently, I feel that the increased level of generality in biomorph formation combined with the extension to the important CaCO₃ system reported in this manuscript justifies publication in Nature Comm. after some minor changes.

1. The authors refer numerous times to the growth mechanism of these structures as assembly of nanorods (“...monohydrocalcite nanorods self-assemble into complex textures...” “They are made of millions of carbonate nanocrystals that are co-oriented via silica-induced interactions ...” “...the carbonate nanocrystals self-arrange...” “... growth takes place by accretion of nanoparticles and not by classical ion-to-ion mechanism.” “The nanoparticles self-aggregate into nanorods ...” “These nanorods spontaneously self-organize ...” The self-assembly mechanism leading to different textures is likely driven by intrinsic anisotropic dipole-dipole interactions between the nanorods.”). Yet there is no evidence given that this is indeed how they form other than the apparent rod-like texture of exposed surfaces in Figs. 4 and S4. Previous literature on these structures has also proposed that the morphology implies nanorod assembly, but, in truth, the formation mechanism is unknown and there is no proof that, even if the rod-like texture is a result of the presence of rod-shaped subunits that are separated by silica coatings, that these units were ever separated in space. Consequently, the authors should not present this mechanism of formation as if it is proven fact.

2. The point that the authors are trying to make with the table of “atom size ratio to Ca” for the various minerals is unclear. The text discusses the fact that only orthorhombic systems have been successfully used, but it says nothing about the size ratio of the metal cation to Ca. However, the table doesn’t indicate which of the minerals are orthorhombic or, for that matter, which ones have been attempted. So there is little connection between the discussion in the text and the table in the figure. One or the other should be modified accordingly.

3. There are a number of typos that need to be corrected. For example, on page 3, “The fail of...” should be “The failure of ... and in Fig. S2, the label that reads “Sear” should be “Sea”.

Reviewer #3 (Remarks to the Author):

The authors demonstrate the formation of biomorphic architectures composed of calcium carbonate mono hydrate (MHC) crystal structure and show that complex shapes can be constructed by controlling the temperature. The work presents an elegant addition to this field, however the manuscript could benefit from a clarification of the role of the temperature (1) and how the MHC phase is formed (2).

(1) The role of role of temperature is described in lines 77-89, and is attributed to modulate the balance between SiO₂ and carbonate precipitation to form a nano composite (e.g. line 126). However no proof of silica deposition is show, and the role of silica is not further addressed. Have the authors established the precipitation of silica, thus forming as claimed a nanocomposite? Also

if the carbonate silica ratio changes as a function of temperature, is there a change in the MHC/SiO₂ ratio as a function of the temperature (for instance in fig 3 &4)? For instance EDAX or TEM on the different segments can give this information

2) Formation of MHC. The experimental procedure in ref. 22 is very similar to the current approach. However they currently obtain only MHC and do not see nucleation of either aragonite or calcite as they previously reported. It would be helpful if the authors clarify how these experiments are different from their previous ones and how this may lead to the exclusive formation of MHC.

Smaller remarks:

Title is not very informative and rather generic: *What/who assists?*

Line 33 Textures: can this term have a clear definition as it reoccurs.

Fig.1. List of Ion radius is not mentioned in the main text. Can this be further explained? Also the list contains minerals that have not been used to form biomorphs, can this be further explained.

Lines 62-72. Please make a clearer distinction between previous & current work.

Line 180: Classical ion-to-ion mechanism: can a reference be added or can this term be explained.

To Reviewer 1:

The authors report a rich self-assembly pathway of trigonal-monohydrocalcite (MHC) based growth of biomorphs as well as spherulitic and filamentary crystals as a function of temperature. The temperature control over the textures and overall morphologies allows for the targeted growth of hierarchical complex structures that modulate the shape and the composition. In my opinion, these are novel results to be published at the level of Nature Communications. The structures are very well characterized given the complexity of the emergent shapes. Before I recommend publication, some minor points mainly with regards to a clearer presentation of the results could be addressed:

1- Lines 69-77 very clearly demonstrate what is new in this manuscript; however, the abstract, although listing novel results, does not clearly communicate the improvements that this manuscript reports over the previous literature. I believe the abstract must be improved along these lines.

Thank you for this suggestion. We have modified the abstract to better emphasize the novelty of the work. The growth of biomorphs is not restricted by crystalline structure, and the finding also opens a promising route to the fabrication of complex multitextured self-assembled material made of silica and chalk.

2- Lines 85-87: Can the authors provide a brief explanation or a qualitative diagram of the anticipated solubility rates of MHC and silica? After all, the statement as it is does not explain why biomorphs do not emerge at 25oC and only spherulites grow.

Answer: We have added a more detailed explanation on the reverse solubility of MHC and silica versus temperature including a plot of solubility versus temperature from 0° to 160 °C, which is included in Supplementary Information (Fig. S1).

Figure S1. Solubility curves of silica and calcium carbonate versus temperature from 0 °C to 160 °C. The solubility curve of amorphous silica is made from the data from the works of Hitchen and Kitahara at alkaline conditions (Ref 1, 2 in Supplementary information); the solubility curve of MHC is made from the data from the work of Kralj et al. at the pH range from 9.7 to 10.5 (Ref 3 in Supplementary information); and the solubility curves of aragonite and calcite are made from the data from the work of Plummer et al. (Ref 4 in Supplementary information).

3- Lines 120-121: The geometry described here is not clear without the aid of visuals: do the authors have a movie or zoomed-in images of Figs. 2 (g-j)? Or a schematic that helps the reader as to what is a “bamboo-like structure,” where are the joints? And arrows depicting the growth vector on these figures?

Answer: We think that the text is explicative but probably the referee is right that it could be enhanced with an illustration. Therefore, we have included in supplementary information a new Figure illustrating this structure (Fig. S6).

Figure S6. FESEM images of curving laminar sheets with bamboo-like joints formed at 60 °C. The general view of the MHC structure is shown in (a); the transition from laminar sheets to branches is shown in (b), the location where the curling occurred is marked by red arrow, and the bamboo-like joints are marked by green arrows; the close-up view of the bamboo-like joint formed along the branch is shown in (c). The morphogenesis of the branches and bamboo-like joints are shown in Fig S5. Scale bars: 200 μm (a), 100 μm (b), 10 μm (c).

4- Line 124: The authors state that the morphologies are controlled by the trigonal symmetry of MHC. However, lines 130-133 and 234-237 oppose that, which I agree with. Perhaps the authors do not see any inconsistency; in any case the statement on Line 124 should be rephrased/clarified.

Answer: We see the keen point of the referee. This line described the shape of MHC at 70 °C, which has the memory of the trigonal crystal structure. But, as we claim and the referee says, the morphology is not controlled by the symmetry. Therefore, we have changed the sentence to say:

They produce single crystals and crystal aggregates with morphologies characteristic of the trigonal symmetry of the MHC crystal structure.

5- Lines 147-149: Can the authors put a reference to the corresponding figures to guide the reader?

Answer: The reference to the corresponding figures have been added in the text.

6- Line 154: The curving laminar sheet is not clear in Fig. 3A, as well as its plane: is the surface normal in the z-axis of the figures?

Answer: Thank you for this comment. Figure 3 contains already a lot of information.

Therefore we have included a new Figure (Fig. S8) in Supplementary Information to explain this thin layer from which the nuclei of the single crystal sticks form.

Figure S8. Panel A: Optical micrographs showing the growth history upon heating from 25 °C to 70 °C (a) and then at constant 70 °C temperature (b, c, d). Panel B: FESEM images of MHC during the transition from 25 °C to 70 °C, the textures of the thin layer and the polycrystalline cluster are shown in close-up views (f, g, h). Scale bars: 100 μm (a, b, c, d), 10 μm (e), 2 μm (f, g, h).

We have also extended the explanation of this layer in the main text as follow:

During the heating, the experiments move across the temperature range 45-60 °C for about 30 minutes, during which the spherulites experience the growth of a polycrystalline thin layer that, in most cases, cover their whole surface, and is the substrate for the formation of polycrystalline clusters. Once the nominal temperature of 70 °C is reached, the growth regime changes and MHC faceted sticks grow from the polycrystalline clusters of the spherical particles (Fig. 3 c and Fig S8).

7- Line 189 and Fig 4a: Is the green region a curved 2D laminar sheet? How do each of the long rods transform into this geometry? Do the authors mean that the valleys within the rods are deposited with silica, and a sheet in turn emerges? Or are there multiple 2D sheets that exhibit a laminar superstructure at the edge-on view; and hence not visible in Fig. 4a? A more detailed explanation and/or some more figures or some guidance (text labels, arrow etc.) added to Fig. 4a would greatly help.

Answer: Thank you for the comment. The sheets do not emerge from the deposition of silica in the valleys within the rods. They form from changing the one-dimensional

growth of the rods into two-dimensional growth to form what we call now blades to describe them better. We have included three additional high magnification FESEM pictures showing the transition from nanorods to blades. We have also made changes in the main text and in the caption of Figure 4.

Figure 4. Heterotextured complexity of MHC formed upon stepwise adjustments of temperature. The growth periods at different temperatures have been labeled with false colors (a): 24 hours growth at 25 °C (yellow), 6 days growth at 45 °C (green), 24 hours growth at 70 °C (red), successively. The close-up views (b-d and f-h) show the transition from the spherulitic core (yellow) grown at 25 °C to the biomorphic laminar sheet (green) grown at 45 °C. The close-up view with higher magnification (d) shows the gradual transition of the textures from micron-size long rods of one hundred nanometer in thickness (25 °C, yellow) to 2D oriented blades (45 °C, green); close-up view (e) shows the mesocrystal with crystallographic symmetry made by nanodrops aggregation on the surface of laminar sheet at 70 °C (red). Close-up views (f-h) shows further details of the transition from nanorods to blades and also show that both the nanorods and the blades are made by accretion of nanodrops. Scale bars: 100 μm (a), 20 μm (b), 2 μm (c, g), 500 nm (d), 1 μm (e, f, h).

8- Line 201: I suppose that the authors meant “nanometers,” not “micrometer.”

Answer: The reviewer is right, it is “nanometers”. We have corrected it in the text.

9- The paragraph starting at line 206: Can the authors put the words yellow, red, green, blue, and pink to the text to enable the reader interpret the figure directly from the text?

Answer: Thank you for the suggestion. We have added the color labels in the main text in addition the caption of Figure 5.

Reviewer #2 (Remarks to the Author):

The manuscript by Zhang et al. reports the growth of biomorphs consisting of CaCO₃ and silica in which complex morphologies are controllably modulated via changes in solution temperature. This is the first report of biomorph growth in a non-orthorhombic crystal system and the first to show that temperature can be used as the control parameter in directing the morphology and in building morphological complexity. Biomorphs represent a particularly significant form of biomimetic structures, because their generation does not require introduction of macromolecular scaffolds, bioorganic additives, or compartmentalization. Rather simple physical controls resulting from coupled precipitation-pH oscillations are all that is required to generate these complex crystalline forms. Consequently, I feel that the increased level of generality in biomorph formation combined with the extension to the important CaCO₃ system reported in this manuscript justifies publication in Nature Comm. after some minor changes.

1. The authors refer numerous times to the growth mechanism of these structures as assembly of nanorods (“...monohydrocalcite nanorods self-assemble into complex textures...” “They are made of millions of carbonate nanocrystals that are co-oriented via silica-induced interactions ...” “...the carbonate nanocrystals self-arrange...” “... growth takes place by accretion of nanoparticles and not by classical ion-to-ion mechanism.” “The nanoparticles self-aggregate into nanorods ...” “These nanorods spontaneously self-organize ...” The self-assembly mechanism leading to different textures is likely driven by intrinsic anisotropic dipole-dipole interactions between the nanorods.”). Yet there is no evidence given that this is indeed how they form other than the apparent rod-like texture of exposed surfaces in Figs. 4 and S4. Previous literature on these structures has also proposed that the morphology implies nanorod assembly, but, in truth, the formation mechanism is unknown and there is no proof that, even if the rod-like texture is a result of the presence of rod-shaped subunits that are separated by silica coatings, that these units were ever separated in space. Consequently, the authors should not present this mechanism of formation as if it is

proven fact.

Answer: Thank you for this interesting point. Actually, the answer to this question depends on how we define self-assembly. There is no doubt that the internal structure of the biomorphs is made by nanorods with a length of ca. 400 nm and a width of 40 nm. The evidence is from transmission and scanning electron microscopy, X-ray diffraction, and atomic force microscopy. As shown in the picture, these independent and co-oriented nanorods can be identified at the growth front, and based on *in-situ* polarized microscopy studies, we know that the co-oriented arrangement of the nanorods occurs during the growth. However, the self-assembly does not occur as could be intuitively expected after formation of the nanorods in the solution and ulterior accretion to the growth front. We have shown that the nucleation and growth of the nanorods takes place in the growth front itself. It was shown that biomorphs with complex morphologies form under stirring of 300 rpm. And a detailed study of the growth front has been reported in Nature Comm. 2017, 8:14427. Therefore, we think that the claim we made is supported by the evidence, so far: the growth front is not a single crystal but a collection of independent nanocrystals that self-arrange keeping co-orientation. This is important because it is the symmetry-breaking of the crystal structure what allows the biomorphs to display such a wonderful panoply of non-crystallographic symmetry.

2. The point that the authors are trying to make with the table of “atom size ratio to Ca” for the various minerals is unclear. The text discusses the fact that only orthorhombic systems have been successfully used, but it says nothing about the size ratio of the metal cation to Ca. However, the table doesn’t indicate which of the minerals are orthorhombic or, for that matter, which ones have been attempted. So there is little connection between the discussion in the text and the table in the figure. One or the other should be modified accordingly.

Answer: Thank you very much for the suggestion. We fully agree with the referee and therefore we have changed Figure 1 by replacing the atom size ratio by the crystallographic structure of the corresponding minerals.

3. There are a number of typos that need to be corrected. For example, on page 3, “The fail of...” should be “The failure of ...” and in Fig. S2, the label that reads “Sear” should be “Sea”.

Answer: Thank you very much. In addition to some authors, several readers have search for typos in the revised manuscript. We have corrected all them.

Reviewer #3 (Remarks to the Author):

The authors demonstrate the formation of biomorphic architectures composed of calcium carbonate mono hydrate (MHC) crystal structure and show that complex shapes can be constructed by controlling the temperature. The work presents an elegant addition to this field, however the manuscript could benefit from a clarification of the role of the temperature (1) and how the MHC phase is formed (2).

(1) The role of role of temperature is described in lines 77-89, and is attributed to modulate the balance between SiO₂ and carbonate precipitation to form a nano composite (e.g. line 126). However no proof of silica deposition is show, and the role of silica is not further addressed. Have the authors established the precipitation of silica, thus forming as claimed a nanocomposite? Also if the carbonate silica ratio changes as a function of temperature, is there a change in the MHC/SiO₂ ratio as a function of the temperature (for instance in fig 3 &4)? For instance EDAX or TEM on the different segments can give this information

Answer: Thank you for this comment. Actually, biomorphs can be properly described as composite non-hybrid but inorganic-inorganic material. They are formed by a crystalline phase of metal carbonate and an amorphous phase of silica. The composition of silica varies with the growth conditions but range from 2% to 7%. The publications reporting the evidence of silica in the formation of biomorphs are cited in the manuscript, in particular Growth behavior and kinetics of self-assembled silica-carbonate biomorphs, Chemistry - A European Journal 2012, 18, 2272-2282. A more detailed and deeper discussion can be found in Nature Comm. 2017, 8:14427. In this manuscript we have included in supplementary information EDX data of Ca/Si ratio showing that the Ca/Si ratio changes with temperature (Fig. S7). Unfortunately, we can not provide data at higher spatial resolution because MHC is extremely unstable under the TEM electron beam.

2) Formation of MHC. The experimental procedure in ref. 22 is very similar to the current approach. However they currently obtain only MHC and do not see nucleation of either aragonite or calcite as they previously reported. It would be helpful if the authors clarify how these experiments are different from their previous ones and how this may lead to the exclusive formation of MHC.

Answer: The growth of aragonite and calcite (Ref. 22) and the growth of MHC are both performed using the same counterdiffusion method (Ref. 22, and also 29, 30 for the growth of MHC), the main difference is the initial concentrations of CaCl₂ and Na₂CO₃: aragonite and calcite are obtained when the initial concentrations is at 0.2 M, while MHC is obtained when the initial concentrations are at 0.05 M. We have added the related information in the text in the introduction part to make it clear.

Smaller remarks:

Title is not very informative and rather generic: What/who assists?

Answer: If the editor agrees, an alternative title could be “Thermal assisted self-assembly of calcium carbonate”

Line 33 Textures: can this term have a clear definition as it reoccurs.

Answer: As the referee knows, in crystallography, crystal structure refers to the organization of atoms in a crystal lattice. Crystal texture refers to the arrangement of crystal within a polycrystalline material. We have introduced a concise definition in that sentence.

Fig.1. List of Ion radius is not mentioned in the main text. Can this be further explained? Also the list contains minerals that have not been used to form biomorphs, can this be further explained.

Answer: We have changed Fig. 1 by replacing the ion radius to the corresponding crystal structure. Since the text discusses more about the crystal structure and crystal morphologies, we think this change can provide a better view to the readers.

Lines 62-72. Please make a clearer distinction between previous & current work.

Answer: Thank you for this suggestion. We have made the text to provide a clearer distinction between the previous works on the growth of aragonite and calcite and the current work on the growth of MHC.

Line 180: Classical ion-to-ion mechanism: can a reference be added or can this term be explained.

Answer: Yes, thank you for the suggestion. We have added a reference from Gal, A., Weiner, S. and Addadi, L., A perspective on underlying crystal growth mechanisms in biomineralization: solution mediated growth versus nanosphere particle accretion. CrystEngComm, 2015, 17, 2606-2615.

REVIEWERS' COMMENTS:

Reviewer #1 (Remarks to the Author):

In my opinion, the authors have satisfactorily addressed all remarks. Therefore, I recommend the publication of the manuscript in Nature Communications.

Reviewer #2 (Remarks to the Author):

The authors have adequately addressed all but one of the comments I provided. I disagree with the response to the comment concerning the use of the term “self-assembly” to describe the growth mechanism. While I continue to strongly support the publication of this work in Nature Comm., I still feel that this issue needs addressing before publication.

I have no problem with the authors description of the growth process in their response: “As shown in the picture, these independent and co-oriented nanorods can be identified at the growth front, and based on in-situ polarized microscopy studies, we know that the co-oriented arrangement of the nanorods occurs during the growth.” However, that does not prove that any assembly takes place. The literal definition of “assemble” (as a transitive verb) is, “to bring together, as in a particular place or for a particular purpose.” The definition of self-assembly is, “a process in which a disordered system of pre-existing components forms an organized structure or pattern as a consequence of specific, local interactions among the components themselves, without external direction.” As stated, these definitions imply that the units were once separated in space and were brought together in an ordered fashion.

The authors often use the term “self-organize,” which is perfectly appropriate, because it does not imply that the individual building blocks were brought together, but rather that the structure arose in an organized manner without the action of external direction, regardless of whether the units formed in place or were assembled after forming. Thus “self-organize” is a very appropriate description of a biomorph and covers all possibilities for the resulting ordered arrangement, but “self-assembly” is only appropriate if movement of the units into place occurred and, whether or not that is indeed how biomorphs form, there is currently no evidence for that process. This is not a matter of semantics, it is a matter of definition.

To further emphasize the point, I will note that if the organized nature of the individual units in a biomorph was an adequate characteristic to justify the term self-assembly, then one could refer to spherulitic growth as a process of self-assembly, because it leads to self-organized structures of individual nanorods or needles. However, that would be a false statement, because each element nucleates in place. Nonetheless the process is one of self-organization, because the location and orientation of each new element is controlled by the existing elements of the structure. Hence the evolution in morphology goes from needles to bundles to dog bones to spherulites.

I do not want to claim that biomorphs form through the process I just described, but, equally, it is wrong to claim they form via assembly. Either may be true or both may be true. No one knows the

answer. Because “self-organization” covers both possibilities, but “self-assembly” only applies to movement of independent units into place, I request that the authors drop the use of the self-assembly and use instead self-organization.

Finally, just so there can be no doubt that the language used in the paper conveys to the reader a process of separately formed nanorods being guided into place, here I reproduce a few of the sentences that I feel clearly convey this:

1) “The transition between textures is rather smooth because in all cases the growth takes place by accretion of nanoparticles and not by classical ion-to-ion mechanism.” This statement is particularly egregious, because, if biomorphs do form by repeated nucleation of new nanorods that form in their final location, then the statements that they accrete and that they do not grow through ion-by-ion processes are both false.

2) “The nanoparticles self-aggregate into nanorods...” This clearly says that particles come together from separated locations.

3) “The self-assembly mechanism leading to different textures is likely driven by intrinsic anisotropic dipole-dipole interactions between the nanorods.” If it turns out that the nanorods nucleate in place, then the dipole-dipole interactions cannot play a role, other than to keep them in place. As stated the sentence implies these forces bring nanorods together.

Reviewer #3 (Remarks to the Author):

The authors addressed my comments and I believe the publication is ready for acceptance.

Dear editor,

Again, thank you very much for managing the editorial process of this manuscript. We are enjoying the review process very much. Here we are presenting our respond to the comments from the referees. For the sake of clarity, the comments of the reviewers are in blue and our answers are in black.

Reviewer #1 (Remarks to the Author):

In my opinion, the authors have satisfactorily addressed all remarks. Therefore, I recommend the publication of the manuscript in Nature Communications.

Reviewer #2 (Remarks to the Author):

The authors have adequately addressed all but one of the comments I provided. I disagree with the response to the comment concerning the use of the term “self-assembly” to describe the growth mechanism. While I continue to strongly support the publication of this work in Nature Comm., I still feel that this issue needs addressing before publication.

I have no problem with the authors description of the growth process in their response: “As shown in the picture, these independent and co-oriented nanorods can be identified at the growth front, and based on in-situ polarized microscopy studies, we know that the co-oriented arrangement of the nanorods occurs during the growth.” However, that does not prove that any assembly takes place. The literal definition of “assemble” (as a transitive verb) is, “to bring together, as in a particular place or for a particular purpose.” The definition of self-assembly is, “a process in which a disordered system of pre-existing components forms an organized structure or pattern as a consequence of specific, local interactions among the components themselves, without external direction.” As stated, these definitions imply that the units were once separated in space and were brought together in an ordered fashion.

The authors often use the term “self-organize,” which is perfectly appropriate, because it does not imply that the individual building blocks were brought together, but rather that the structure arose in an organized manner without the action of external direction, regardless of whether the units formed in place or were assembled after forming. Thus “self-organize” is a very appropriate description of a biomorph and covers all possibilities for the resulting ordered arrangement, but “self-assembly” is only appropriate if movement of the units into place occurred and, whether or not that is indeed how biomorphs form, there is currently no evidence for that process. This is not a matter of semantics, it is a matter of definition.

To further emphasize the point, I will note that if the organized nature of the individual units in a biomorph was an adequate characteristic to justify the term self-assembly, then one could refer to spherulitic growth as a process of self-assembly, because it leads to self-organized structures of individual nanorods or needles. However, that would be a false statement, because each element nucleates in place. Nonetheless the process is one of self-organization, because the location and orientation of each new element is

controlled by the existing elements of the structure. Hence the evolution in morphology goes from needles to bundles to dog bones to spherulites.

I do not want to claim that biomorphs form through the process I just described, but, equally, it is wrong to claim they form via assembly. Either may be true or both may be true. No one knows the answer. Because “self-organization” covers both possibilities, but “self-assembly” only applies to movement of independent units into place, I request that the authors drop the use of the self-assembly and use instead self-organization.

Finally, just so there can be no doubt that the language used in the paper conveys to the reader a process of separately formed nanorods being guided into place, here I reproduce a few of the sentences that I feel clearly convey this:

1) “The transition between textures is rather smooth because in all cases the growth takes place by accretion of nanoparticles and not by classical ion-to-ion mechanism.” This statement is particularly egregious, because, if biomorphs do form by repeated nucleation of new nanorods that form in their final location, then the statements that they accrete and that they do not grow through ion-by-ion processes are both false.

2) “The nanoparticles self-aggregate into nanorods...” This clearly says that particles come together from separated locations.

3) “The self-assembly mechanism leading to different textures is likely driven by intrinsic anisotropic dipole-dipole interactions between the nanorods.” If it turns out that the nanorods nucleate in place, then the dipole-dipole interactions cannot play a role, other than to keep them in place. As stated the sentence implies these forces bring nanorods together.

Reviewer #3 (Remarks to the Author):

The authors addressed my comments and I believe the publication is ready for acceptance.

We are very grateful to the comments from referee 1 and 3, their suggestions undoubtedly help us a lot to improve this manuscript.

We also thank referee 2 for her/his very valuable comments and discussion in this submission. The criticism maintained by referee 2 is interesting indeed. We think that the distinction between self-organization and self-assembly is not anymore defined by etymology. Today, the meaning of self-organization is different within different disciplines. This is also true for self-assembly. In a paper entitled “Consisted concepts of self-organization and self-assembly” (Complexity, 2008 DOI 10.1002/cplx.20235), Halley and Wrinkler claimed that both terms describe processes that give rise to collective order from dynamic small-scale interactions. They also highlight the complex nature of the boundary between these processes with some examples and stated: “Unfortunately, the term self-assembly is sometimes used interchangeably with the term self-organization and other words and phrases of imprecise or multiple meaning. The

use of the concept of self-assembly is increasing in many disciplines, with a different flavor and emphasis in each.”

We were very cautious using the terminology in previous papers on silica-carbonate biomorphs to avoid any misunderstanding on what we claim, to make clear that the biological emulation is constrained to morphology and mineral texture, as well as to the nature of the mineral precipitation process. So far, we have not used the term self-organization because it could be considered a term pretentious in some context, particularly for biologists. For a good bunch of biologists (see for instance Abel and Trevors, *Physics of Life Reviews* 3 (2006) 211–228) the inanimate matter can only self-order but not self-organize. However, in the context of physics and physico-chemistry self-organization would be very well accepted. Actually, in a recent publication, Nakouzi & Steinbock (2016, *Science Advances* 2 : e1601144) classified silica-carbonate biomorphs as self-organized structures.

Therefore, if the editor agrees, we are happy to use self-organization instead of self-assembly. We have replaced “self-assembly” by “self-organization” and “self-assembled” by “self-organized” through the whole paper. However, on page 11 we keep the term self-assembly because there we are describing specifically the mechanism of formation of nanorods by accretion of nanoparticles.

Looking forward to your news.

Best regards,

Juanma García-Ruiz
5th November 2018